# Feature Selection and Molecular Classification of Cancer Phenotypes: A Comparative Study

**DOI:** 10.3390/ijms23169087

**Published:** 2022-08-13

**Authors:** Luca Zanella, Pierantonio Facco, Fabrizio Bezzo, Elisa Cimetta

**Affiliations:** 1Department of Industrial Engineering (DII), University of Padova, 35131 Padova, Italy; 2Fondazione Istituto di Ricerca Pediatrica Città della Speranza (IRP), 35127 Padova, Italy

**Keywords:** feature selection, classification, learning algorithm, cancer, gene expression

## Abstract

The classification of high dimensional gene expression data is key to the development of effective diagnostic and prognostic tools. Feature selection involves finding the best subset with the highest power in predicting class labels. Here, we conducted a comparative study focused on different combinations of feature selectors (Chi-Squared, mRMR, Relief-F, and Genetic Algorithms) and classification learning algorithms (Random Forests, PLS-DA, SVM, Regularized Logistic/Multinomial Regression, and kNN) to identify those with the best predictive capacity. The performance of each combination is evaluated through an empirical study on three benchmark cancer-related microarray datasets. Our results first suggest that the quality of the data relevant to the target classes is key for the successful classification of cancer phenotypes. We also proved that, for a given classification learning algorithm and dataset, all filters have a similar performance. Interestingly, filters achieve comparable or even better results with respect to the GA-based wrappers, while also being easier and faster to implement. Taken together, our findings suggest that simple, well-established feature selectors in combination with optimized classifiers guarantee good performances, with no need for complicated and computationally demanding methodologies.

## 1. Introduction

While traditional diagnoses and prognoses are built on a combination of clinical and physical examination and medical history, research is increasingly relying on in silico procedures based on high-throughput gene expression data to detect disease [1,2]. Feature selection is a crucial process in the fast growing fields of pattern recognition and machine learning [3], while classification is the supervised learning task of predicting the categories of new observations on the basis of training data. The goal of performing feature selection on gene expression data is thus typically twofold: class prediction, and biomarkers identification [4]; here, we will focus on the first.

Typical classification tasks are to separate healthy from cancer patients based on specific gene expression signatures or to discriminate among different types of tumors [4,5]. In the machine learning jargon, they correspond to solving a classification problem with gene expression data as an input. Although the field is now moving toward sequencing-based methods, a conspicuous number of existing data in the ArrayExpress [6] and GEO [7] databases, are from microarray platforms [8].

Microarray analysis, adopted for large-scale interrogation of gene expression, uses microchips with immobilized labeled DNA probes combined with imaging and data processing to quantify specific hybridization [9]. Feature selection and classification can however be affected by typical characteristics of microarray data such as: i. large dimensionality up to several tens of thousands of genes, with small sample size; ii. possible class imbalance especially in multiclass datasets, and iii. dataset shift, frequent when observations come from different experiments and leading to a change in distribution of features or class boundaries in data collection or dataset splitting [10,11,12]. Taken together, they impact error estimation and can lead to the generation of unsubstantiated hypotheses [13]. As a provocatory example, a reanalysis of 7 major published microarray-based studies aimed at predicting cancer prognosis, revealed that 5 out of 7 did not classify patients better than chance [13].

As in our case when features are genes, another general issue lowering pattern recognition performance is the high-dimensional feature vectors characterizing these samples, often containing intra-class variability generating noise and irrelevant information [14,15]. Feature pre-processing typically improves the accuracy of classification by removing intra-class natural variability [16]. Specifically, feature selection aims at finding the subset of data, that guarantees the best class prediction performance and: i. reduces the computational demand by building faster and more cost-effective predictors; ii. increases classification accuracy and iii. improves results clarity [17]. Overall, gene selection allows for a better monitoring of target diseases and a deeper understanding of the processes that generated the data [17].

Feature selectors can be divided into three broad categories: i. filters, ii. wrappers and iii. embedded approaches [18,19]. Filters select subsets of features using the general characteristics of the data, independently on the classification learning algorithm. Given their good generality and low computational complexity, they are suitable for high-dimensional datasets [20], but do not provide subsets that are finely tuned for the downstream classifier. The wrapper paradigm uses the predictive accuracy of the learning algorithms as a black box to score subsets of features according to their predictive power [21]. The generality of features they provide is limited, at the expense of a higher computational demand [22], a reduced robustness to parameter changes in the classification algorithms [18], and tendency to overfit [23]. The main advantage is that they produce feature sets finely tuned for the coupled classification learning algorithm, in principle guaranteeing better performance. The embedded methodology incorporates feature selection in the training process and is usually specific to given learning machines [21]. Examples include decision trees or artificial neural networks. Hybrid methods employing a filter to reduce the dimensionality of the search space spanned by a wrapper are also emerging.

The objective of this paper is to evaluate the efficiency in classification of several combinations of feature selection and learning algorithms on gene expression data. Since feature selection is intrinsically classifier-dependent, as each learning algorithm has different sensitivity to changes of the feature space [24], we employed 4 feature selectors in combination with 5 classification learning algorithms. Such algorithms exploit different conceptual architectures for class prediction and broadly cover the spectrum of the most used methods in the machine learning literature. To evaluate the potential impact of the dimensionality of input data with respect to the predictive performance of each classifier, we generated gene sets of different sizes. Predictive performances were evaluated on three benchmark microarray datasets: SMK_CAN_187 [25], GLI_85 [26], and CLL_SUB_111 [27].

While the literature is teeming with methods for feature selection and classification, we believe that our contribution will help the reader understand that, although there is no unique combination performing best across all datasets, a tailored approach evaluating the characteristics of the data and the appropriate algorithms can lead to significant results without excessive computational weights.

## 2. Results

The chosen feature selectors are: 1. minimum redundancy maximum relevance (mRMR), 2. Relief-F, 3. Chi-squared, and 4. Genetic Algorithm (GA); combined with the following classification learning algorithms: 1. Random Forests (RF), 2. Partial Least Squares-Discriminant Analysis (PLS-DA), 3. Support Vector Machines (SVM), 4. Regularized Logistic Regression (RLR), and 5. k-Nearest Neighbors (kNN). These feature selectors and classification learning algorithms were chosen as instances of different conceptual architectures (see Section 4 for details).

To verify if our results are independent on the specific dataset split, we computed our metrics over 100 resamplings when using classifiers trained on mRMR, Relief-F and Chi-squared. This number of resampling was suggested by sensitivity analysis, as large enough to ensure stable predictions. With GA-based wrappers, we based our evaluations on a single test set due to the higher computational demand.

### 2.1. Predictive Performance

In evaluating the predictive performance of each combination of feature selector and classifier, we will first thoroughly analyze the accuracy, and then briefly address all other metrics. Our results are then compared with those reported in the literature.

#### 2.1.1. Accuracy Analysis

Figure 1 shows the predictive accuracy on SMK_CAN_187 using a subset of 5, 10, 20, 40 and 50 genes. With all feature selectors, PLS-DA guarantees accuracies in the range 90–100%, ~20% higher than all other learning algorithms. Independently of the feature selection method, RF, SVM, RLR and kNN yield comparable results, and the 95% confidence interval (CI) computed over the resamplings is typically smaller than the differences between methods (all data are available in Appendix A). Overall, we observe that the choice of the selector does not impact accuracy, suggesting that for small gene sets (0.01–0.03% of the whole data) all selectors tested give features relevant to the target classes without adding noise to the classification. Moreover, when using GA-based wrappers, increasing the population size from 50 to 200 individuals and the maximum number of generations from 35 to 150 does not improve the accuracy (results not shown, see Appendix A).

We then ranked the features and selected genes sets of increasing size (20, 30, 40, 50, 60, 70 and 80% of the whole SMK_CAN_187 dataset); in doing so, we omitted GA-based wrappers due to their computational demand. Figure 2 shows that PLS-DA still guarantees better performances (>90%), while the other classifiers stratify with decreasing accuracy from PLS-SA to RLR, SVM, RF, and kNN, independently of the filter applied.

This suggests that the choice of the learning algorithm has greater impact on a successful classification when the problem is tackled in the high-dimensional domain.

Figure 3 and Figure 4 show the predictive accuracies on GLI_85 using small and large gene sets, respectively, and confirm that the choice of the feature selection algorithm does not impact the accuracy of the classifiers, and that increasing the population size and the number of generations does not provide substantial improvements (Appendix A). In this case however, all classifiers provide comparable accuracies in the range 77–100% (please see Appendix A for further details), with no learning algorithms outperforming the others. A neater stratification is established when using large feature sets, with SVM surpassing the other learning algorithms independently of the filter it was combined with.

Finally, Figure 5 and Figure 6 give predictive accuracies on the ternary CLL_SUB_111 dataset. From Figure 5 we observe a common trend towards improved average accuracy with increasing numbers of retained features, possibly due to the multi-class nature of the dataset and its class unbalance. Again, there are only minor differences attributable to the choice of feature selector, and large increases in the population size and the number of generations does not provide appreciable improvements (Appendix A). Figure 6 shows that all classifiers yield comparable performance independently of the filter and number of features retained, except for kNN which yields 20% lower average accuracies. Although there is no unambiguous explanation for such behavior, we recall that kNN strongly relies upon the concept of distance: as the number of features increases, the distance between closest points grows exponentially, approaching the average inter-point distance. This translates into a gradual “loss” in the reliability of nearest neighbors’ computation, ultimately lowering kNN predictive capabilities. We argue that complicated classification tasks, such as multiclass-unbalanced problems, could make such limitation more severe.

#### 2.1.2. Other Metrics Analysis

Although accuracy provides a useful means to compare classifier, it can be biased in presence of unbalanced datasets such as CLL_SUB_111. We thus further compared our combinations of feature selectors and classifiers based on recall, precision, specificity, NPV and F_1_ score (or their macro-averages in case of multiclass problems). All data are available in the Appendix A; we will here discuss the most significant highlights.

As for accuracy, the value of each metric is largely dataset dependent. For instance, PLS-DA provides the best recall, precision, specificity, NPV and F_1_ score in combination with each of the filters when classifying the SMK_CAN_187 dataset, in some cases outperforming other learning algorithms by almost 30%. However, this is not always true when considering GLI_85 where precision and specificity by PLS-DA are 10% lower compared to the other classifiers, or CLL_SUB_111 where all methods yield comparable results. As an example, Table 1 gives the average ±95% CI of each metric for all classifiers on the datasets when employing a set of 20 features selected with Relief-F. In most cases, the best performing classifier is not only the one providing the highest average value for the given metric, but also the most stable across resamplings. Overall, these metric values are in good agreement with those reported for the predicted accuracy. While this does not hold true in general, and especially in multiclass cases (CLL_SUB_111), it supports the use of predictive accuracy as the central metric for discussion and comparison also in the following section.

#### 2.1.3. Comparison with Literature Results

Table 2 summarizes reported predictive accuracies resulting from a literature survey on the datasets.

For SMK_CAN_187 we found a range between 47 and 83%, depending on the algorithm and number of selected features evaluated on regular and stratified 5-fold cross validation studies [4]. Nematzadeh at al. report accuracies in the range 59–71% with SVM, Naïve Bayes (NB) and single decision trees (DT) without a preliminary feature selection step [28]. On average DT led to lower accuracies compared to our RF, stable at ~70% in combination with all filters; conversely, DT seemed to benefit from a feature selection step [29]. Newly proposed feature selection methods were also tested in combination with kNN, DT, NB and SVM, reporting accuracies of ~70% [16,28]. Remarkably, our RLR, RF and PLS-DA appear more suitable for the classification of this dataset, with PLS-DA outperforming all other learning algorithms by ~20% in most combinations, irrespective of the number of features and the feature selector. While in some cases these differences might be partially affected by the slightly larger sample size that we employed in the training phase (85% of the original dataset vs. 80% in [28]), we exclude that they could be biased by the specific random splits, as we averaged the performance over 100 resamplings. Additionally, RF tended to outperform DT, suggesting that ensemble versions of the methods can lead to improvements in the results, as recently shown [30,31].

Predictive accuracies for the GLI_85 dataset are in the range 71–92%, with a tendency towards improved performance by SVM, both alone and combined with feature selectors [16,28,32]. GA-based wrappers coupled with different classifiers (DT, NB and SVM) led to maximum accuracies of 80% [32]. These results are consistent with our study, and we again notice an overall increase in the predictive performance of RF over DT.

For CLL_SUB_111, reports give predictive accuracies below 80% with less than 100 features and using a ranked selection of nearest discriminant features [16,33]. Our results are comparable to those reported in literature as far as small gene sets are employed as predictors. We noticed remarkable improvements when employing larger sets of features with PLS-DA, suggesting that predictive accuracy benefits from the feature extraction mechanism at the core of the NIPALS algorithm exploited in PLS-DA.

Although not exhaustive due to the vast existing literature, these comparisons give an insight into some general trends. Interestingly, well-known feature selection algorithms guarantee results comparable to those achieved by methods designed ad hoc (e.g., [28]). Moreover, in two cases PLS-DA allowed peerless performance, possibly due to its inherent feature selection process through latent variables construction which allows easier handling of *small n–large p* datasets. Last, RFs provide a substantial improvement over DTs, which works in favor of the use of the ensemble paradigm as an effective tool for building classification learning algorithms.

**Table 2 ijms-23-09087-t002:** Summary of classification accuracies attained by different combinations of feature selectors and classification learning algorithms as reported by selected works.

Dataset	Accuracy (%)	Feature Selectors	Classification Learning Algorithms	Reference
**SMK_CAN_187**	47–83%	no FS INT mRMR CFS	C4.5 DT Naïve Bayes SVM	[4]
	59–71%	no FS Whale [28]	SVM Naïve Bayes DT	[28]
	66–71%	mRMR IG [29,34] Ranked features selection	SVM Naïve Bayes kNN	[16]
**GLI_85**	71–85%	no FS IG Relief-F FCBF	C4.5 DT NB SVM-RFE kNN	[16]
	78–92%	no FS whale	SVM NB DT	[28]
	72–80%	GA-based wrappers	DT NB SVM	[32]
**CLL_SUB_111**	60–80%	mRMR IG Ranked features selection	kNN SVM NB	[16,33]

#### 2.1.4. Comparison with Variance-Based Unsupervised Feature Selection

To further extend our investigation encompassing alternative methods for gene prioritization, we studied how the aforementioned selection techniques compare with filtering based on ranking highly variable features. Variance-led gene prioritization can serve as a means to perform feature selection in an unsupervised manner, by retaining those features that exhibit high variance across data without the need to stratify samples according to class membership *a priori*. Such an approach is straightforward and motivated by the fact that the variance of a gene should reflect its biological heterogeneity.

Following the same experimental design as in the previous section, we trained classification learning algorithms on 100 resamplings of each dataset and evaluated predictive performance on the classification of SMK_CAN_187, GLI_85 and CLL_SUB_111. Again, to test the impact of both small and large feature sets on the downstream classifiers, we generated feature sets of 5, 10, 20, 40, 50 and 20, 30, 40, 50, 60, 70, 80% of the whole gene set. The resulting Accuracy, Recall, Precision, Specificity, NPV and F_1_ score are summarized in the available Appendix A.

When dealing with large feature sets (Appendix A) training classifiers on subsets of genes ranked by variance yields comparable performance to those attained using mRMR, Relief-F, Chi-Squared and GA-based wrappers, for all the classification learning algorithms trained. For the binary datasets, SMK_CAN_187 and GLI_85, when resorting to few biomarkers (Appendix A) variance-based gene selection yields classifiers with both lower Accuracy and F_1_ score when inputting five to ten genes. The predictive outcomes tend to improve by employing larger number of features (20, 40, 50 genes). Analogously, when tackling multiclass classification (CLL_SUB_111) variance-based gene selection yields both lower Accuracy and lower values of the other metrics with respect to both supervised filters and GA-based wrappers in the whole range of 5–50 genes, a behavior that can be ascribable to the imbalance of this dataset, where one of the classes is largely unrepresented (10% vs. 44 and 46%).

#### 2.1.5. Runtime Comparisons

Runtime, the elapsed time during the execution of a program, is a major factor in the assessment of an algorithm and a comprehensive investigation cannot prescind from addressing this topic. As such, we evaluated the runtime required for feature selection and training of each classification learning algorithm. For each classification learning algorithm, we also performed hyperparameter optimization, as will be better described in the following sections. Table 3 reports the runtimes collected by running a test on the SMK_CAN_187 dataset. Classifiers were trained on 50 features. Analogously, GA-based wrappers were aimed at generating the best subset of 50 genes. Showing runtimes (instead of CPU times) is convenient to provide a practical indication of the actual time and is more insightful for algorithms exploiting parallelization. All tests were conducted on a PC running Windows 10 Enterprise OS, version 21H2 (Build: 19044.1826), equipped with 64 GB RAM and a 6 cores thread Intel^®^ Core^TM^ i7–8700 K CPU with clock speed of 3.70 GHz. GA-based optimization was run in parallel (six workers). Each runtime refers to a single run of the given algorithm. Due to differences in algorithmic implementations, coarseness of the employed grids in hyperparameter optimization (e.g., *T* in RF and is *A* in PLS-DA are optimized across 256 and 15 values, respectively) and different parallelization architectures (when exploited), we partially discourage comparing runtimes for different classification algorithms. Conversely, comparing the same classification algorithm with and without GA-based wrapper is far more meaningful. Time needed for filtering-based feature ranking is very short, in the order of seconds, with the sole mRMR taking 3 min, because the algorithm scales quadratically with the number of features and the dataset is high-dimensional [35]. Noticeably, GA-based wrappers yield a substantial increase in the runtime for all classifiers. The result is more apparent for some algorithms such as: i. RF, where the runtime increases from (B) 10 s (without GA-based wrapper) to (C) 44 min and (D) 2 h (with GA-based wrappers of 50 and 200 individuals, respectively); ii. SVM, from (B) 2 s to (C) 1 h and (D) 13 h, and iii. even for kNN, for which the runtime increases by (C) 300 and (D) almost 800 times with respect to (B), despite its simplicity and the limited range [1,2,3,4,5,6,7,8,9,10,11,12,13,14,15] in which *k* was optimized. Again, it is worth remarking that each time corresponds to a single run and that we advise running several resamplings/bootstraps (as in this study) to attain stable outcomes. Moreover, one might consider searching a larger parameter space, which increases the computational demand of each generation of a GA-optimization. A similar behavior is expected for an increased number of features. Last, rows (C) and (D) in Table 3 show the strong impact of increasing the population size (and the number of generations) in a GA-optimization routine. Therefore, for challenging tasks, or to simply span a larger number of features combinations which demands a larger population at every GA generation, one must deal with a substantial increase in the computational demand.

## 3. Discussion

Our study evaluates the impact of several combinations of feature selectors and learning algorithms on the molecular classification of cancer by gene expression profiling on three benchmark microarray datasets. We employed three well-known filters: Chi-squared, mRMR and Relief-F, and a GA-based wrapper, with five classifiers: SVM, RF, RLR/RMR, PLS-DA and kNN. To evaluate the performance of each selector/classifier we computed accuracy, recall, precision, specificity, NPV and F_1_ score on a test set (20% holdout). We generated feature sets of different size to evaluate whether the dimensionality of the input space could affect the outcomes of the classifiers. Our results reveal that once a given classification learning algorithm is set: *1.* all of the filters achieve comparable performance on the same dataset both when considering few features (5–50) and large gene sets (20–80% of the whole set); *2.* filters achieve comparable or even better results in terms of all of the metrics employed with respect to the GA-based feature selectors, although the latter are much more computationally demanding and finely tuned for each classifier. *3.* the predictive performance achieved on small and large feature sets are comparable, and most classification learning algorithms can handle the large dimensionality of this type of data quite well. The only exception to this behavior is related to the use kNN on CLL_SUB_111.

For the same feature selector, we also compared the outcomes of the different learning algorithms. We conclude that in most cases different learning algorithms: *1.* guarantee similar performance on the same dataset, the only exceptions being kNN and PLS-DA which under- and outperform all the others in two cases, respectively; *2.* the datasets themselves set an upper bound to the predictive performance since while successful class prediction can be easily achieved for some dataset independently of the selector/classifier employed, improving the outcomes is a challenge for others.

Last, we compared our results with published literature highlighting that: *1.* linear classifiers work quite well for the classification of microarray data, and *2.* ensembles of decision trees such as Random Forests, typically outperform single DT, suggesting that the ensemble paradigm is beneficial to improve performance.

Taken together these results suggest that, as a first order effect, the quality of the data relevant to the target classes is the key aspect for a successful classification of cancer phenotypes. Moreover, the specific combination of feature selection and classification learning algorithms plays a secondary role, with most combinations yielding similar overall performance. We found very few exceptions to this rule on the analyzed data, both when considering few features and large gene sets. Last, we observed that simple filters tend to guarantee comparable or better results than those provided by the computationally demanding GA-based wrappers.

Interestingly, while many observed that selecting a small number of discriminative genes from thousands is essential for successful sample classification [36], we showed that feature selection does not improve, on average, the predictive performance of five widely employed classifiers. Conversely, the downside is also true: in most cases, we showed that when it comes to realizing a diagnostic test, very few informative genes (5–50) are sufficient for class prediction. Last, we show that linear classifiers are in general good enough to obtain satisfactory performance with the advantage of a higher interpretability. If it is true that accuracy generally requires more complex prediction methods and that simple and interpretable functions do not make the most accurate predictors [37], we argue in favor of the Occam’s razor, claiming that successful prediction performance on microarray data can be attained also with relatively simple and interpretable classifiers.

To conclude, while research proceeds in developing new methods to improve the results, our study suggests that simple, well-established feature selectors coupled with optimized classifiers can guarantee satisfactory performance, without the need to resort to highly demanding methodologies.

## 4. Materials and Methods

### 4.1. Classification Learning Algorithms

Given a set of features $x=\left(x_{1},\ldots ,\;x_{p}\right)$, the task of learning from observations (or samples) is to find an approximate definition for an unknown function f(x) given training examples in the form $\left(x_{i},f\left(x_{i}\right)\right)$. Once a dataset is fixed, the classification learning algorithm is the general methodology used to learn a specific classifier, the function that maps an input feature space to a set of class labels (or targets) [38]. The classifiers considered in this work are:**RF**: Random Forests (RF) consist of a combination of a large number of tree predictors, *T*, each voting for a class, m=1, …,M [37]. A bootstrap sample of the training set and a random selection of the input features generate each tree using both bagging [39] and random feature selection. Each decision tree gives a class prediction based on the samples features and gets a vote; the class with the most votes is the forest prediction. RF are not prone to overfitting, even when employing a large number of features, require minimal data cleaning and are effective in variance reduction, while exhibiting good interpretability [40].**PLS-DA**: Partial Least Squares-Discriminant Analysis (PLS-DA) is based on the PLS regression algorithm [41] where the dependent variable represents class membership. The method constructs *A* latent variables, i.e., linear combinations of the original features, by maximizing their covariance with the target classes. The number of latent variables *A*, accounting for the dimensionality of the projected space, is the hyperparameter to be optimized in the training phase. Being inherently based on a dimensionality reduction step, PLS-DA is fit for high-dimensional data as it allows locate and emphasize group structures when discrimination is the goal and reduction is needed [42]. Last, the interpretation of PLS-DA models in terms of the contribution of each feature to class discrimination is eased by the analysis of weights (**W**) and loadings (**P**) [41], making PLS-DA a versatile tool for predictive and descriptive modelling in biomedical applications and diagnostics [42,43].**SVM**: Support Vector Machines (SVM) [23] is a popular tool for binary classification. SVM seeks the decision boundary that separates all data points of the two classes, i.e., the one with the largest margin between the two classes. The training data is a set of observation-label pairs, and the equation of the hyperplane separating the classes is the solution of an optimization problem. The optimality is influenced only by points that are closer to the decision boundary and thus more difficult to classify. SVM is effective and memory efficient in spanning high-dimensional spaces, as it uses only the support vectors (a subset of the training points) in the decision functions, favoring its application in many areas of bioinformatics [23].**RLR:** Regularized Logistic Regression (RLR) measures the relationship between a categorical class label and one or more features, by estimating probabilities using a logit link function [44]. Here we employed a L1-regularization, robust to irrelevant features and noise in the data [45], and fitted the penalized maximum-likelihood coefficients by solving the optimization problem in (A8) (see Section A.1). When generalizing to the multiclass case *M* > 2, we employed multinomial logistic regression conceived as *M*–1 independent binary logistic models, where the predicted class is the one with the highest score. Additional information is reported in Section A.1. Simple to perform, logistic regression is particularly useful when it comes to model categorical variables, commonly used in biomedical data to encode a set of discrete states (as in the present work). RLR also allows predicting class-associated probability [46] and further enhances model interpretability, particularly favoring applications in high-dimensional domains [44].**kNN**: *k*-Nearest Neighbors (kNN) is an instance-based method [47]. Given a matrix **X** of *n* observations and a distance function, a kNN search finds the *k* observations in **X** that are closest to a (or a set of) query samples. The number of nearest neighbors, *k*, is the main hyperparameter to tune. In general, small values of *k* lead to classifiers with weak generalization abilities, while large *k*’s increase the computational demand. kNN assigns labels to new unknown samples using labels of the *k* most similar samples in the training set, based on their distance to observations (Euclidean in this work). Despite its simplicity, kNN is a suitable choice even when data have nonlinear relationships, making it a benchmark learning rule to draw comparisons with other classification techniques [48].

### 4.2. Feature Selectors

Feature selection can be formulated as a combinatorial optimization problem where the objective function is the generalization performance of the predictive model, usually quantified by an error, and design variables are the inclusion or the exclusion of the input features. Exhaustive feature selection would evaluate 2p different combinations, where *p* is the dimensionality of the features space, making it unfeasible for high-dimensional datasets and thus requiring improved methods [49]. We here detail the characteristics of the chosen feature selection algorithms: three filters and a more computationally demanding wrapper based on Genetic Algorithms.

#### 4.2.1. Filters

**mRMR**: the minimum redundancy maximum relevance (mRMR) selects features that are minimally redundant (maximally dissimilar to each other), while being maximally relevant to the target classes [50]. mRMR requires binning, and bins reflect the number of values assumed by each gene. Redundancy and relevance are quantified using the pairwise mutual information of features and mutual information of feature and target classes, respectively [22]. Relevant features are necessary to build optimal subsets, given their strong correlation with the target classes, and the mRMR algorithm performs a sequential selection with a forward addition scheme. At each iteration, the feature that is mostly relevant to the target and least redundant compared to the already selected ones is added [20].**Relief-F**: Relief-F ranks the most informative features using the concept of nearest neighbors by weighting their ability to discriminate observations under different classes using distance-based criteria functions [22]. Features assigning different values to neighbors of different classes are rewarded, while those that give different values to neighbors of the same class are penalized [4]. The algorithm samples random observations and locates their nearest neighbors from the same and opposite classes. The values of the features of the nearest neighbors are compared to the sampled instance and used to update the relevance scores for each feature. Relief-F is sensitive to features interaction without evaluating combinations of features [51], making it computationally efficient (its complexity is *O*(*n*^2^*p*), where *n* is the number of observations), yet sensitive to complex patterns of associations. Unlike mRMR, Relief-F does not remove redundant features, but adding features that are presumably redundant can lead to noise reduction and better class separation, as very high variable correlation (or anti-correlation) does not exclude variable complementarity [21]. Relief-F is widely used because of its simplicity, high operation efficiency and applicability to multiclass problems [52].**Chi-squared**: an individual Chi-squared test evaluates the relationship between each feature and the target classes: when they are independent, the statistics takes on a low value (high *p*-value), while high values indicate that the hypothesis of independence between the feature and the target class can be rejected (small *p*-value). Since Chi-squared test works for categorical predictors and gene expression data are continuous variables, we binned the expression values of each gene (10 bins) before each test. Unlike mRMR and Relief-F, Chi-squared is a univariate technique and evaluates each feature independently of the others.

#### 4.2.2. Genetic Algorithms for Feature Selection

Genetic Algorithms (GA) are a heuristic optimization method inspired by evolution theory and comprise probabilistic search procedures designed to work on large spaces involving states that can be represented by strings [53]. GA repeatedly modify a population of individual solutions using a distributed set of samples from the space (a population of strings). Leardi et al. [54] claim that the subsets of variables selected by GA are generally more efficient than those obtained by classical methods, since they produce a better result while using a lower number of features.

In the GA framework, we define an *individual* as any set of features to which an objective function is applied, and a *population* as an array of individuals. One of the crucial choices when using GA is the population size, which corresponds to the number of *possible solutions* for the optimization problem. Smaller populations result in faster search exploration with lower accuracy, while larger populations increase the accuracy at the expense of a higher computational demand. A typical GA scheme follows different steps (Appendix A): 1. Initialization; 2. Fitness assignment, highlighting individuals with higher probability of being selected; 3. Selection; 4. Crossover, recombining selected individuals to generate a new population; 5. Mutation, adding a source of variation, randomly changing some of the included features in the offspring. Steps 2–5 are repeated until a stopping criterion is satisfied. The solution to the process is the best individual, characterized by its set of features. While classical optimization algorithms generate a single point at each iteration and select the next point in sequence by a deterministic computation, GA generate a population of points and selects the next population by random recombination of the features.

Additional details are available in Appendix B.

### 4.3. Workflow

Figure 7 summarizes the workflow for features selection through filters. To evaluate the predictive performance of feature selection techniques in combination with classifier learning algorithms, we randomly split each dataset into training and test set, with a 15% holdout. Features were ranked using mRMR, Relief-F and Chi-squared on the remaining 85%. Differently from mRMR and Chi-Squared, Relief-F requires the number of nearest neighbors *k* to be known, so we set *k* = 10 as suggested in the literature [51]. From the ranked set of features, we built sets of incremental sizes composed of 5, 10, 20, 40, and 50 genes, covering fractions in the range 10^−4^–10^−3^ of the original datasets. To test whether the dimensionality of input data affects the outcomes of the classifiers, we also generated *large* feature sets comprising 20, 30, 40, 50, 60, 70 and 80% of the entire gene set. This procedure was repeated 50 times, each one resampling from the original dataset using the Mersenne Twister pseudo-random number generator varying the initialization seed between 1–50 for reproducibility. This shuffling makes the feature selection process more robust to data splitting.

When building GA-based wrappers, we cast our problem as a constrained optimization for each classifier with parameters settings shown in Table 4 and solved the optimization for population sizes of 50 and 200 individuals and two maximum number of generations, 35 and 150. The population was initialized by ranking features by class separability criteria using a filter, and then randomly sampling from the pool of the most informative genes. We then employed the ‘mutation adapt feasible’ routine available in MATLAB (Natick, MA) [55], and a scattered procedure as a crossover rule generating a random binary vector and selecting genes corresponding to the logical 1 from the first parent, and 0 from the second to form the child. The fraction of individuals undergoing crossover was set as 8% of the total, while the *elites*, individuals that guarantee suboptimal values of the objective function, but are still passed to the next generation to guarantee diversity in population, was set to 5%.

We trained each classification learning algorithm on 85% of the original data. Hyperparameter tuning resulted from a 10-fold cross-validation for all classifiers but RF, where we used the out-of-bag prediction error [56]. Whenever possible, we generated stratified partitions to decrease class imbalance among the folds. Table 5 presents the hyperparameters, their ranges of variation, and objective functions optimized in the training phase of each learning algorithm. The optimal set of hyperparameters was the one that minimized the misclassification rate. Briefly, the optimal number of trees, *T*, was the one yielding the minimum out-of-bag prediction error. For PLS-DA we optimized the number of latent variables *A* by varying it between 1–15, avoiding increasing it further to reduce the risk of overfitting. Latent variables were constructed with the NIPALS algorithm [57]. PLS1 and PLS2 were used for binary and multiclass problems, respectively [58]. For our linear SVM, model selection was achieved by tuning the value of *C* across 20 logarithmically-spaced points in the range 10^−5^–10^3^ to span different degrees of penalty for the soft-margin formulation. In multiclass problems, we extended SVM using a *one*-vs-*one* coding design, i.e., by training *M*(*M−1*)/2 binary SVMs, where *M* is the number of unique class labels and kept the same range for *C*. For RLR and the corresponding multiclass counterpart RMR, we optimized λ by generating a geometric sequence of coefficients in the range 0–$\lambda_{\max}$, where $\lambda_{\max}$ is the largest value that yields a non-null classifier. In training kNN, we varied *k* between 1–15, by considering only odd values to break ties. Euclidean distance was used to quantify the similarity between the set of data and the query points. Features were standardized before running the algorithms to avoid those in greater numeric range dominating the others. Finally, we estimated predictive performance on test sets and averaged the results of each resampling. All computational experiments were coded using MATLAB R2020a and R2020b (The MathWorks, Natick, MA, USA). For the multinomial logistic regression, we used the glmnet package [59].

### 4.4. Datasets

We employed three widely used medical datasets (Table 6), benchmarks in the domain of feature selection and classification.

SMK_CAN_187 derives from a diagnostic study on lung cancer [25]; the dataset is binary (smokers with and without cancer), contains expression profiles of 19,993 genes measured across 187 samples and has low class imbalance.

The GLI_85 dataset comes from a large-scale gene expression analysis on 85 diffuse infiltrating gliomas [26]; it is again binary but exhibits higher class imbalance.

CLL_SUB_111 was collected from a microarray-based subclassification of patients affected by B-cell chronic lymphocytic leukemia [27]. This dataset comprises three classes and has even higher imbalance.

### 4.5. Performance Assessment Criteria

To assess the performance of the combined approaches, we evaluated the following metrics: accuracy, recall, precision, specificity, negative predictive value (NPV) and F_1_ (Equations (1)–(6)). The accuracy of a classifier (1) identifies the percentage of correctly classified positive and negative samples. Recall (or sensitivity) (2) assesses the fraction of observations correctly classified as positive (TP) out of all true positives (TP + false negatives (FN)), and specificity (4) the correct negative (TN) out of all true negatives (TN + false positives (FP)). Similarly, precision (3) and NPV (5) report the fraction of observations correctly classified as positive, or negative, out of all the observations that were predicted as positive, or negative. For a binary classification problem, recall, specificity, precision and NPV provide the same information assuming that two classes have been exchanged. Last, the F_1_ score (6) is the harmonic mean between precision and recall, favoring classifiers that achieve similar precision and recall.
(1)$$\mathrm{Accuracy}\;=\frac{\;\mathrm{Number}\;\mathrm{of}\;\mathrm{correct}\;\mathrm{predictions}\;}{\;\mathrm{Total}\;\mathrm{number}\;\mathrm{of}\;\mathrm{predictions}\;}$$
(2)$$\mathrm{Recall}=\frac{\mathrm{TP}}{\mathrm{TP}+\mathrm{FN}}$$
(3)$$\mathrm{Precision}=\frac{\mathrm{TP}}{\mathrm{TP}+\mathrm{FP}}$$
(4)$$\mathrm{Specificity}=\frac{\mathrm{TN}}{\mathrm{TN}+\mathrm{FP}}$$
(5)$$\mathrm{NPV}=\frac{\mathrm{TN}}{\mathrm{TN}+\mathrm{FN}}$$
(6)$$F_{1}\;\mathrm{score}=2\;\frac{\mathrm{Precision}\;\cdot \;\mathrm{Recall}}{\mathrm{Precision}\;+\;\mathrm{Recall}}\;(0\leq F_{1}\leq 1)$$

In the case of multiclass classification (CLL_SUB_111), we macro averaged each *per class* metric as follows:(7)$$\mathrm{Metric}=\frac{1}{M}\sum_{m=1}^{M}{\mathrm{Metric}}_{j}$$
where Metric denotes any macro-average of the metrics (2)–(6), suffix *j* indicates the *per class* metric and *M* is the number of classes. This approach treats each class equally, without giving more weight to heavily biased classifiers [60].

## 5. Conclusions

The classification of high dimensional gene expression data is key to the development of effective diagnostic and prognostic tools. In this study, we evaluated different combination of feature selectors and classification learning algorithms on cancer-related datasets. Our findings suggest that, although the distinct conceptual architectures underpinning different classification learning algorithms play a role in predictive performance, advantages of one method with respect to another are often highly context specific. Conversely, the quality of the data relevant to the target classes is always the key for a successful classification of cancer phenotypes. As such, how to best combine specific feature selection and classification learning algorithms *a priori* cannot be determined. Despite this caveat, our study consistently shows that the use of simple filters for feature selection typically provides sufficiently informative feature sets, even when the size of the generated gene sets is relatively low (e.g., 5) and that in most cases there is no need to resort to computationally demanding wrappers. We believe that our work will be instrumental in aiding scientists in the conceptualization and design of studies that involve feature selection and classification, a broad domain characterized by an ever-growing number of available methods, which is often central in molecular biology and biomedicine. The guidelines presented will thus be effective in helping scientists to make decisions in this context.

## Figures and Tables

**Figure 1 ijms-23-09087-f001:**
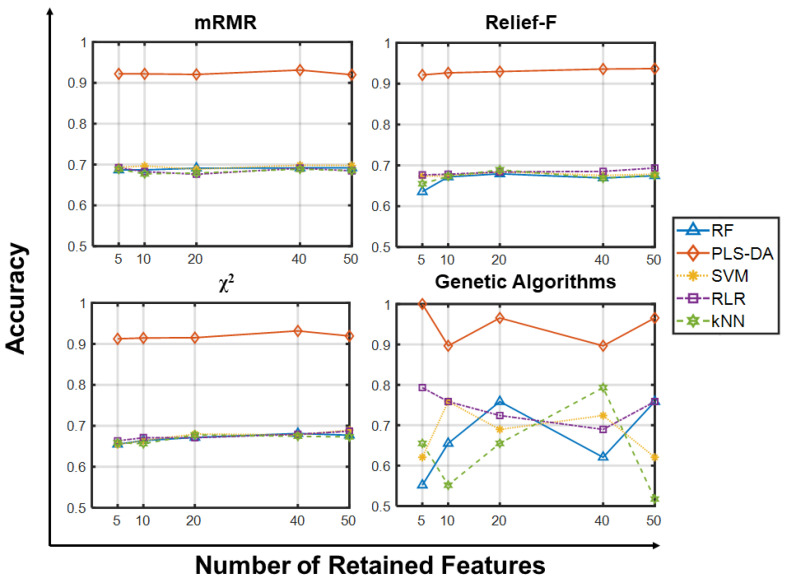
Predictive accuracy using few retained features on SMK_CAN_187. Predictive accuracies of the classifiers (RF, PLS-DA, SVM, RLR, kNN) vs. Number of Retained Features (5, 10, 20, 40, 50) on the SMK_CAN_187 dataset using the four feature selection methods. Each point in the panels for mRMR, Relief-F and Chi-squared is the average predictive accuracy computed over 100 resamplings, while those for GA corresponds to a single value.

**Figure 2 ijms-23-09087-f002:**
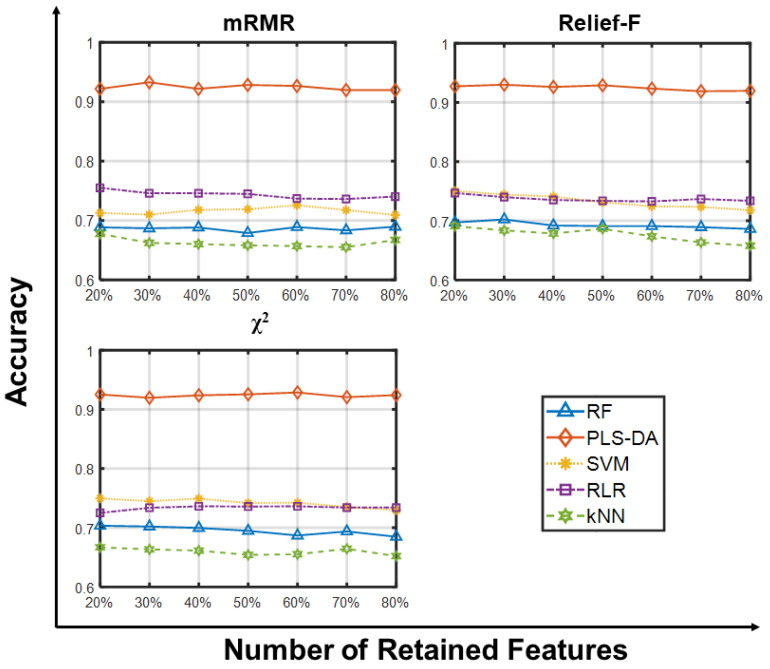
Predictive accuracy retaining larger sets of features on SMK_CAN_187. Predictive accuracies of RF, PLS-DA, SVM, RLR and kNN vs. Number of Retained Features (20, 30, 40, 50, 60, 70, 80% of the whole gene set) on the SMK_CAN_187 dataset using three feature selection methods. Each point is the average predictive accuracy computed over 100 resamplings.

**Figure 3 ijms-23-09087-f003:**
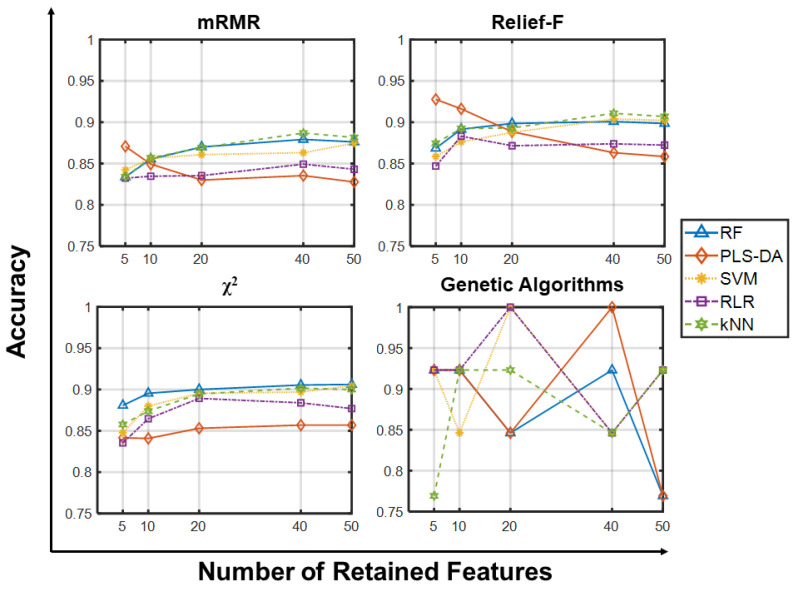
Predictive accuracy using few retained features on GLI_85. Predictive accuracies of the classifiers (RF, PLS-DA, SVM, RLR, kNN) vs. Number of Retained Features (5, 10, 20, 40, 50) on the GLI_85 dataset using the four feature selection methods. Each point in panels for mRMR, Relief-F and Chi-squared is the average predictive accuracy computed over 100 resamplings, while those for GA corresponds to a single value.

**Figure 4 ijms-23-09087-f004:**
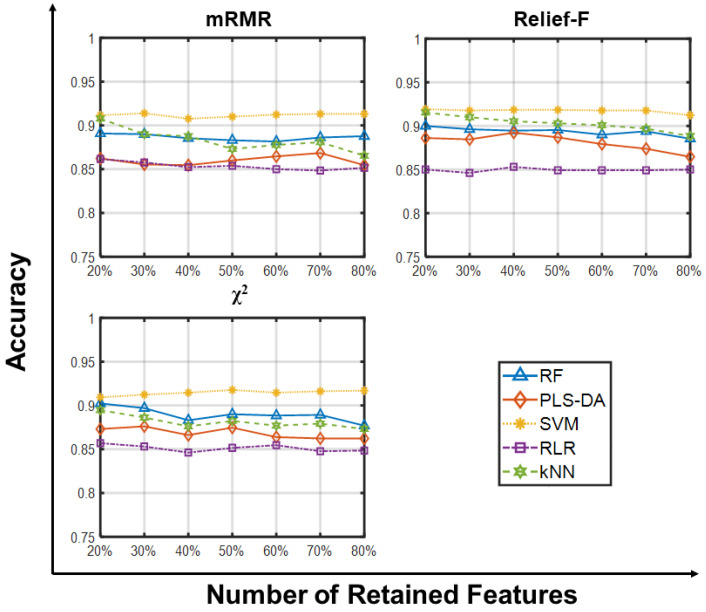
Predictive accuracy retaining larger sets of retained features on GLI_85. Predictive accuracies of RF, PLS-DA, SVM, RLR and kNN vs. Number of Retained Features (20, 30, 40, 50, 60, 70, 80% of the whole gene set) on the GLI_85 dataset using three feature selection methods. Each point is the average predictive accuracy computed over 100 resamplings.

**Figure 5 ijms-23-09087-f005:**
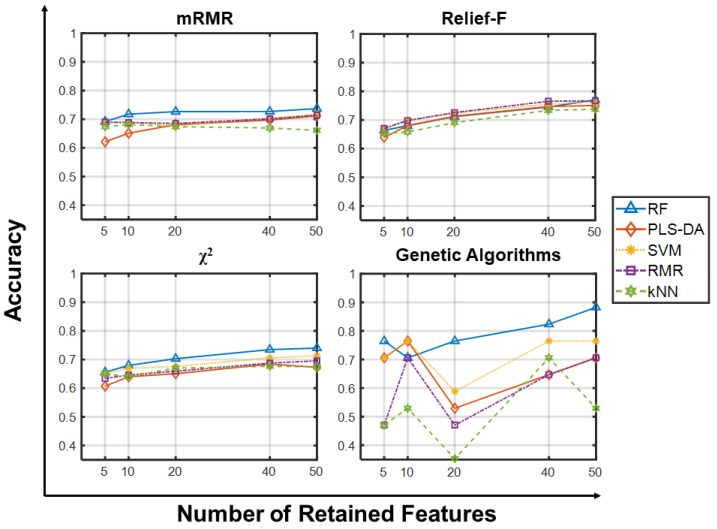
Predictive accuracy using few retained features on CLL_SUB_111. Predictive accuracies of the classifiers (RF, PLS-DA, SVM, RLR, kNN) vs. Number of Retained Features (5, 10, 20, 40, 50) on the CLL_SUB_111 dataset using the four feature selection methods. Each point in panels for mRMR, Relief-F and Chi-squared is the average predictive accuracy computed over 100 resamplings, while those for GA corresponds to a single value.

**Figure 6 ijms-23-09087-f006:**
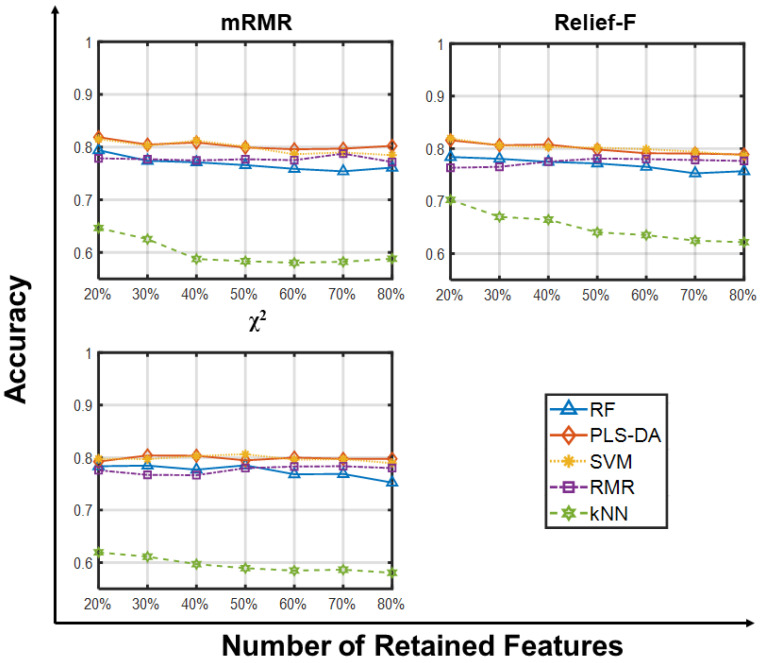
Predictive accuracy retaining larger sets of retained features on CLL__SUB_111. Predictive accuracies of RF, PLS-DA, SVM, RLR and kNN vs. Number of Retained Features (20, 30, 40, 50, 60, 70, 80% of the whole gene set) on the GLI_85 dataset using three feature selection methods. Each point is the average predictive accuracy computed over 100 resamplings.

**Figure 7 ijms-23-09087-f007:**
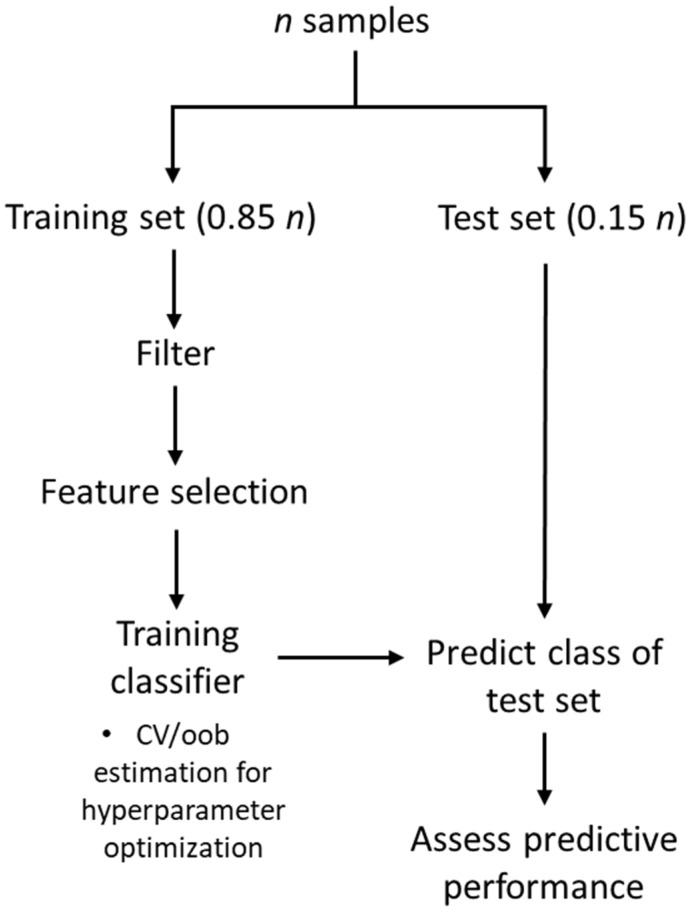
Workflow employed for filter-based feature selection and classification. CV = cross-validation; oob = out-of-bag.

**Table 1 ijms-23-09087-t001:** Average ± 95% confidence interval (%) assumed by each metric. Classification was performed using RF, PLS-DA, SVM, RLR/RMR and kNN on sets of 20 features selected using Relief-F. Results were computed over 100 resamplings. Best values for each combination of classifier/dataset are underlined.

		Accuracy	Recall	Precision	Specificity	NPV	F_1_ Score
	**RF**	67.93 ± 1.58	61.24 ± 2.58	67.6 ± 2.01	73.72 ± 2.09	69.08 ± 1.62	63.55 ± 1.99
	**PLS-DA**	92.97 ± 1.05	93.53 ± 2.02	92.97 ± 1.73	92.43 ± 2.02	95.29 ± 1.4	92.46 ± 1.15
**SMK_CAN_187**	**SVM**	68.41 ± 1.61	60.14 ± 2.56	69.37 ± 2.32	75.6 ± 2.41	68.9 ± 1.51	63.52 ± 2.04
	**RLR**	68.38 ± 1.59	65.73 ± 2.56	66.83 ± 2.01	70.63 ± 2.26	70.96 ± 1.78	65.59 ± 1.85
	**kNN**	68.9 ± 1.57	60.07 ± 2.31	70.03 ± 2.17	76.55 ± 2.17	69.04 ± 1.49	64 ± 1.85
	**RF**	89.85 ± 1.51	74.5 ± 4.37	87.3 ± 3.26	95.48 ± 1.14	91.54 ± 1.45	78.36 ± 3.42
	**PLS-DA**	88.85 ± 1.73	100 ± 0	73.92 ± 3.31	84.81 ± 2.31	100 ± 0	83.92 ± 2.23
**GLI_85**	**SVM**	88.77 ± 1.58	73.58 ± 4.34	84.82 ± 3.66	94.46 ± 1.35	91.04 ± 1.43	76.71 ± 3.48
	**RLR**	87.15 ± 1.63	73.92 ± 4.64	79.19 ± 3.78	91.94 ± 1.53	91.18 ± 1.47	74.7 ± 3.44
	**kNN**	89.31 ± 1.63	70.83 ± 4.63	88.42 ± 3.47	96.2 ± 1.13	90.29 ± 1.51	76.6 ± 3.74
	**RF**	71.18 ± 1.78	78.32 ± 1.38	78.45 ± 1.52	82.7 ± 1.09	83.35 ± 1.09	77.36 ± 1.43
	**PLS-DA**	71.29 ± 1.97	78.71 ± 1.46	75.58 ± 2.11	83.3 ± 1.13	83.65 ± 1.13	75.64 ± 1.87
**CLL_SUB_111**	**SVM**	72.41 ± 1.89	79.4 ± 1.45	80.28 ± 1.55	83.35 ± 1.14	84.1 ± 1.14	78.91 ± 1.5
	**RMR**	72.59 ± 1.85	78.59 ± 1.57	80.6 ± 1.54	83.49 ± 1.12	84.41 ± 1.12	78.3 ± 1.54
	**kNN**	69.06 ± 2.02	77.01 ± 1.5	77.34 ± 1.71	81.35 ± 1.22	82.09 ± 1.22	76.2 ± 1.59

**Table 3 ijms-23-09087-t003:** Runtime to (**A**) rank features with each feature selector, (**B**) train classification learning algorithms on 50 features, (**C**) solving the GA-based optimization problem with 50 features, population Size = 50 and maximum number generations = 35, and (**D**) solving the GA-based optimization problem with 50 features, population size = 200 and maximum number of generations = 150. Values are reported in hours, minutes, seconds and milliseconds (HH:mm:ss.SSS).

**(A) Feature Selection**	**mRMR**	**Relief-F**	**Chi-Squared**	**Variance**	
runtime	00:03:23.420	00:00:08.355	00:00:00.074	00:00:00.038	
**(B) Classifier Training**	**RF**	**PLS-DA**	**SVM**	**RLR**	**kNN**
runtime	00:00:10.517	00:00:00.322	00:00:02.041	00:02:41.384	00:00:01.518
**(C) GA-based wrapper** 50—Pop Size:50, Max Gen: 35	**RF**	**PLS-DA**	**SVM**	**RLR**	**kNN**
runtime	00:44:49.874	00:00:51.536	01:10:54.360	00:42:12.297	00:05:06.008
**(D) GA-based wrapper** 50—Pop Size:200, Max Gen: 150	**RF**	**PLS-DA**	**SVM**	**RLR**	**kNN**
runtime	02:06:01.370	00:01:26.498	13:06:10.923	10:11:14.675	00:13:46.126

**Table 4 ijms-23-09087-t004:** Settings for the optimization using Genetic Algorithms.

Option	Setting
Population size	50–200
Max Generations	35–150
Mutation Function	‘mutation adapt feasible’
Crossover Function	crossover scattered
Crossover Fraction	8%
Selection Function	‘selectionstochunif’ [55]
Elite count	5% of Population Size

**Table 5 ijms-23-09087-t005:** Ranges for the hyperparameters of the classification learning algorithms and settings for the optimization using Genetic Algorithms.

	RF	PLS-DA	SVM	RLR/RMR	kNN
**Hyperparameter**	*T*	*A*	*C*	*λ*	*k*
**Range**	1–256	1–15	10^−5^–10^3^	$0-\lambda_{\max}$	1–15
**Objective function**	out-of-bag error	cv-error	cv-error	deviance	cv-error

**Table 6 ijms-23-09087-t006:** Datasets employed in our comparative study.

	Samples	Features	Number of Classes	Class Distribution	Reference
**SMK_CAN_187**	187	19,993	2	48–52%	[25]
**GLI_85**	85	22,283	2	31–69%	[26]
**CLL_SUB_111**	111	11,340	3	10–44–46%	[27]

## Data Availability

The data underlying this article are available in: https://jundongl.github.io/scikit-feature/datasets.html (accessed on 10 August 2022) and can be freely accessed; GEO at https://www.ncbi.nlm.nih.gov/geo/ (accessed on 10 August 2022) and can be accessed with accession ID GSE4115, GSE83294 and GSE2466. All manuscript data are available in: http://researchdata.cab.unipd.it/id/eprint/679 (accessed on 10 August 2022).

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
