# Peer review of "Feature Selection and Molecular Classification of Cancer Phenotypes: A Comparative Study"

_ijms, 2022, doi:10.3390/ijms23169087_

Round 1
Reviewer 1 Report
Authors sufficiently addressed all my concerns and ready for publication after editorial guidelines are met.
Author Response
We sincerely thank the Reviewer for endorsing our publication. We are truly grateful for the opportunity that we have been given to publish our work on the International Journal of Molecular Science.
Reviewer 2 Report
Although the paper is well organized, the following comments should be reflected on the paper.
1. To obtain stable results, please increase the number of resampling to 100 at least.
2. Although the paper is a comparative study of the feature selector and ML combination, the number of overlapped gene names in each (5,10,30,40,50) for the combination of methods in the appendix or the paper for biologists.
3. The results with GA should be compared. please show the running time results for each feature sector and ML combination.
4. Feature selector and ML should be specified - for example, linear algorithm is used in SVM or the number of K is ...(K is specified) 5. In figure7, cv to select the optimal method of combination, should be added. 6. contents such as feature selectors in the appendix are copied from the body of the paper. duplicate contents should be deleted. 7. recent references should be dominant. minor comments- On page 12, any a set of features -> any set (grammar)
In conclusion, it is worthwhile to publish in the journal if the authors review the paper in detail and then revise it. (The journal IF is so high)
Author Response
Although the paper is well organized, the following comments should be reflected on the paper.
We thank the Reviewer for the in-depth observations and the valuable comments on our manuscript; we carefully addressed all of them. All suggestions helped us greatly improve the quality of our work and the clarity of the manuscript. We hereby present responses to all the comments.
- To obtain stable results, please increase the number of resampling to 100 at least.
We thank again the Reviewer for this comment. We did take the impact of the number of resamplings into account when designing our study and, specifically, conducted a sensitivity analysis to assess the impact of the number of resamplings on the stability of the outcomes provided by several classification learning algorithms. Sensitivity analysis revealed that all metrics used for performance assessment did not exhibit significant differences after 40-50 resamplings. This result motivates our choice of 50 resamplings.
The trend is evident in the Supplementary Tables where all metrics exhibit very narrow confidence intervals (CI): the 95% CI for the mean accuracy is always <4% for any combination of dataset/FS/classification learning algorithm, with the 95% CI for the other metrics being just slightly above 4% in very few cases. To give a better idea and provide a more detailed response to the reviewer, we here attach an additional plot showing the Accuracy and F1 score of the 5 classification learning algorithms vs the number of features (5, 10, 20, 40, 50) for the SMK CAN 187 dataset. The left and right column show results averaged over 50 and 100 bootstraps, respectively. Feature sets for the specific figure were generated using the most variable genes, since this was the only feature selection scheme that we had not tested when conceptualizing the study. Noticeably, differences between the corresponding plots are minimal and with no statistical significance. We further arranged mean and 95% CI values for all the inspected metrics into a table, SensitivityAnalysis_test.xlsx: the comparison with SupplementaryTables_Variance rev.xlsx, which reports the same calculation over 50 resamplings, in general highlights that only minimal differences can be related to the number of resamplings.
To further clarify this point, we explicitly motivated the choice of 50 resamplings in the body of the manuscript (Results, page 3), as a number that guarantees stable results for fair comparisons.
- Although the paper is a comparative study of the feature selector and ML combination, the number of overlapped gene names in each (5,10,30,40,50) for the combination of methods in the appendix or the paper for biologists.
The Reviewer raised a good point. However, the data from the database we used were already preprocessed and did not contain information about the given gene name (just a generic annotation in the form gene1, … , geneN). As such, we could not investigate how many of the selected genes were known markers to the relevant cancer type/subtype and we decided to focus on the computational aspects of our analysis.
- The results with GA should be compared. please show the running time results for each feature sector and ML combination.
We totally agree with the Reviewer: estimation of running times is key to exhaustive algorithmic comparisons. We now included section 2.1.5 in the main manuscript body, that shows the running time of feature selectors, training of classification learning algorithms and GA-based wrappers. We clearly see that GA-based wrappers substantially increase the runtime with respect to the sole classifier training and that the sum of the time needed to perform feature selection using a filter and to train the classifier is substantially shorter than the time needed to train the GA-based wrapper. Results are included in Table 5.
- Feature selector and ML should be specified - for example, linear algorithm is used in SVM or the number of K is ...(K is specified)
We thank again the Reviewer for the comment. We indeed specified feature selectors and classifiers in our work.
- Feature selectors: for Relief-F, the k parameter was set as k=1, as stated in section “4.3 Workflow” at page 13, which is typical value suggested in the literature for similar problems (obtained from benchmarking).
- Classifiers: since each learning algorithm has different sensitivity to changes of the feature space, we always performed hyperparameter optimization, i.e. whenever training the classification learning algorithm, both on a predefined feature set or within a GA-based wrapper. Please refer to Table 5 for the hyperparameter ranges of each algorithm: T (Random Forests) 1-256, A (PLS-DA) 1-15, C (SVM) range 10-5-103, lambda (Regularized Logistic/Multinomial Regression) range 0-lambda_max (please see main text for description of how lambda_max is determined), k (kNN) in 1-15.
We agree that the previous caption of the Table was a bit misleading, and it looked like if those hyperparameters ranges only referred to the GA-optimization problem. For the sake of clarity, we updated and improved the caption.
- In figure7, cv to select the optimal method of combination, should be added.
In agreeing with the Reviewer in highlighting this point, we added the “CV/oob estimate for hyperparameter optimization” to Figure 7.
- contents such as feature selectors in the appendix are copied from the body of the paper. duplicate contents should be deleted.
Thank you again for this comment and for helping us improve the readability of the manuscript. We removed unnecessary duplicate contents from the Appendix to limit redundancies. As such, we:
- Removed all the redundancies in Appendix A.1 – Classification Learning Algorithms. We only left the mathematical description of the methods, i.e. the main equations for RF, PLS-DA, SVM and RLR/RMR. We also removed the description of kNN, since the material provided in the body of the text is explanatory enough.
- We renamed Appendix A.2 as – “Genetic Algorithms for Feature Selection” and removed any description regarding filters, which was indeed equivalent to the one provided in the body of the manuscript. We centered the content of the appendix on the description of the main steps employed in the GA-based optimization, that is only briefly introduced in “Materials and Methods”.
- recent references should be dominant. minor comments
As suggested by the Reviewer, we removed 2 outdated references and, to align the manuscript to the state-of-the-art in feature selection and classification, we surveyed the literature and added 7 references to papers published in the period 2020-2022 in high-IF journals. These papers are relevant to the field and enrich the discussion of our manuscript; they specifically deal with:
- classification of microarray data (Vahid et al 2022 In: Comput Biol Med; Rostami et al 2022 In: Artif Intell Med);
- ensemble methods in feature selection and classification (Vahid et al 2022 In: Comput Biol Med; Zhu et al 2021 In: Knowl Based Syst);
- classification of cancer subtypes (Pozzoli et al 2020 In: Artif Intell Med; Mahin et al 2022 In: Genomics);
- the use of PLS-DA in diagnostics (Hao et al 2022 In: Acta A.);
- Relief-based methods for feature selection (Zhang et al 2022 In: Knowl Based Syst).
- On page 12, any a set of features -> any set (grammar)
The typo was corrected, we also carefully checked our manuscript and improved the overall quality of the language.

Round 2
Reviewer 2 Report
I think most of my comments were reflected. However, the result of 100 times resamplings was not given to the paper. I believe that the paper should include the result of at least 100 times of resampling to increase the quality of the paper for IJMS.
Author Response
Authors response to the Reviewers comments
First of all, we thank the Editor and the Reviewer for the opportunity of submitting a revised version of our work. We tracked changes in red font color in the revised manuscript, following the Reviewer suggestion.
Reviewer 2:
I think most of my comments were reflected. However, the result of 100 times resamplings was not given to the paper. I believe that the paper should include the result of at least 100 times of resampling to increase the quality of the paper for IJMS.
We thank the reviewer for the comment. According to the suggestion we now re-run all the codes with 100 resamplings and consequently updated all results. The new figures in the main manuscript and Supplementary Figures.docx show the average predictive accuracy computed across 100 resamplings. Analogously, Table 1 in the main manuscript and tables in Supplementary Tables.xlsx report the average values and 95% CI computed across 100 resamplings. The newly generated figures resemble the previous ones, i.e. those with 50 resamplings: this was expected, as suggested by our previous sensitivity analysis (see the answers to the previous round of reviews). However, increasing the number of resamplings to 100 resulted in even narrower confidence intervals.
In thanking the Reviewer, we confirm that we thoroughly checked the manuscript and included all the updated results. We are confident that this round of reviews has further improved the quality of the manuscript for IJMS.

Round 3
Reviewer 2 Report
It is worthwhile to being published on the journal .